# Advancing Few-shot Continual Learning via Selective Knowledge Transfer

## Abstract

Continual learning with language models (LMs) is a promising and challenging research that greatly impacts many applications. Existing solutions treat previous tasks equally, making them vulnerable to task interference, lacking scalability with a large number of tasks, and oblivious to the intrinsic relationships among tasks. This work presents *selective knowledge transfer (SKT)*, a novel framework towards continual learning with LMs. SKT aims to maximize positive knowledge transfer while systematically minimizing the effects of irrelevant information from dissimilar tasks. To this end, SKT first assesses the degree of interference between the current and previous tasks and then selectively aggregates the tasks that maximize knowledge transfer for continual training. In addition, we integrate SKT into the current state-of-the-art continual language learning algorithm, Progressive Prompts, to introduce *Log-evidence Progressive Prompts (LePP)*, which facilitates knowledge transfer between tasks. Comprehensive evaluations on challenging few-shot continual learning benchmarks demonstrate that LePP can surpass existing baselines for continual learning with LMs with minimal overhead. Our extensive ablation studies reveal that SKT can discover useful task correlations without any prior knowledge, many of which align with human evaluations. Code will be published upon acceptance.

## 1 Introduction

Modern language models are required to efficiently adapt to dynamic environments with only a few labeled samples, to enable widespread adoption in the real world. This adaptability allows users to effectively interact with language models, *e.g.* a chatbot, by providing a couple of input-output pairs to achieve their desired outcomes. This goal can be achieved through Few-shot Continual Learning (FSCL) (Zhang et al., 2024b; Pasunuru et al., 2021; Li et al., 2022), where the language models can adapt to non-stationary distributions in a few-shot manner. However, two main obstacles arise in learning a task stream: preventing catastrophic forgetting (CF), where the model's performance on previous tasks significantly deteriorates after learning a new one, and encouraging forward transfer (FT), which leverages learned knowledge to facilitate the learning process of new tasks.

Existing approaches (Kirkpatrick et al., 2017; Rusu et al., 2016; Nguyen et al., 2017; Yoon et al., 2018) address forgetting by isolating model parameters related to past knowledge. For example, in *regularization-based approaches* (Kirkpatrick et al., 2017; Nguyen et al., 2017; Huang et al., 2021), the model parameters of future tasks are constrained to remain close to those of previous ones by adding auxiliary regularizers to the final loss function. Besides, *parameter isolation-based approaches* (Rusu et al., 2016; Yoon et al., 2018) allocate new parameters for each new task and freeze the neurons associated with prior tasks to preserve knowledge. This method mitigates forgetting and leverages prior knowledge for FT through parameter sharing.

Research on continual learning (CL) for NLP tasks has extended these ideas to enable language models to continually learn new tasks without forgetting. In particular, the NLP community has focused heavily on the parameter isolation-based approach (Ke et al., 2020; 2021; Razdaibiedina et al., 2023; Zhang et al., 2024b; Peng et al., 2024) because learning with pre-trained LMs significantly improves the performance of CL systems and LMs are flexible to adapt to new tasks with parameter-efficient fine-tuning (PEFT) (Houlsby et al., 2019; Hu et al., 2021; Lester et al., 2021; Li & Liang, 2021). We argue that existing *PEFT-based approaches* are vulnerable to task interference due to

their naive aggregation mechanisms (Razdaibiedina et al., 2023), and they lack scalability because of inefficient similar task selection procedures (Peng et al., 2024; Ke et al., 2020; 2021). Therefore, these methods hinder the possibility of positive forward transfer, which plays a crucial role in FSL where learning from similar tasks leads to better performance (Zhou et al., 2021), and the system scalability with the number of tasks.

In this study, we propose *selective knowledge transfer* (SKT), a novel and generalized framework to maximize the positive forward transfer of LMs in FSCL settings. In particular, our framework consists of two stages: (1) **Selection** where relevant memories are chosen; (2) **Aggregation** where the correlated past memories are effectively aggregated to facilitate the adaptation of the current task. Following these principles, we devise *Log-evidence Progressive Prompts (LePP)*, a CL algorithm utilizing transferability measures (TMs) to select similar tasks. Unlike previous works that require learning each task's representation as a task key or a probe soft prompt, LePP only necessitates a single forward pass over the few-shot dataset for each trained prompt, making it more computationally efficient than its counterparts.

To validate the effectiveness of *LePP*, we conduct experiments on several challenging continual NLP benchmarks. Experimental results demonstrate that *LePP* outperforms existing CL approaches with prompt tuning. By integrating selective knowledge transfer (SKT) into its mechanism, LePP effectively leverages relevant past knowledge while discarding irrelevant information, thereby accelerating the learning of the current task and showcasing its scalability with an increasing number of tasks in a sequence. Additionally, our ablation studies reveal that the task correlations identified by our proposed framework align with human evaluations.

In summary, our paper presents three significant contributions:

- We introduce *selective knowledge transfer* (SKT), a novel CL framework for LMs. Our framework leverages TMs, a recent advancement in model selection, to enable systems to autonomously identify memories relevant to the current task. To our best knowledge, this is the first exploration of applying TMs in CL with LMs.

- We integrate our proposed framework into current prompt-based CL algorithms that can maximize positive knowledge transfer to improve system performance.

- We conduct extensive experiments to demonstrate the superiority of our proposed framework over previous SoTAs on popular NLP benchmarks. Ablation studies reveal that *SKT* can uncover task correlations within a stream, which helps to understand the effect of task relatedness on the performance of the CL system. In addition, we show that *SKT* can work with different data modalities including images.

## 2  PRELIMINARIES

**Continual learning**    Task Incremental Learning (TIL) consecutively trains a neural network $f_\theta$ on a sequence of datasets $\mathcal{T} = \{T_1, T_2, \ldots, T_M\}$, where M is the number of datasets in the sequence. $T_t = \{x_i^t, y_i^t\}_{i=1}^{N_t}$ contains $N_t$ annotated inputs $(x_i^t, y_i^t)$ drawn from an unknown distribution $\mathbb{P}_t(X, Y)$ i.e., $(x_i^t, y_i^t) \sim \mathbb{P}_t(X, Y)$. Here, $X$ is the space of input $x$, while $Y$ is the space of label $y$ with $p$ classes, where $|Y| = p$. In Few-shot Learning (FSL), we sample $k$ sentences per class for each task as inputs. In our TIL setting, during training at time step $t$, the model is prohibited from accessing the previous data $T_{i<t}$ due to privacy issues, and the task-ids are available during inference.

**Prompt tuning**    Prompt Tuning (PT) (Lester et al., 2021) attaches a trainable token $P = [p_1, p_2, \ldots, p_l] \in \mathbb{R}^{l \times d}$ at the beginning of the input sequence's embedding $X = [e_1, e_2, \ldots, e_n] \in \mathbb{R}^{n \times d}$ of an input $x$, where $n$ and $l$ are the number of tokens in the input sequence $x$ and the prefix $P$, respectively. The prompted input $\hat{X} = [P, X] \in \mathbb{R}^{(l+n) \times d}$ is then forwarded to the network $g_\theta$, which is initialized with pre-trained weights $\theta_0$ to maximize the log-likelihood of $Y$. Formally, for each task $T$, we optimize the soft prompt $P$ while keeping the model's weight unchanged: $P^* = \arg\max_P \sum_{(x,y) \in T} \log p(y|x, P, \theta_0)$.

Inspired by Progressive Neural Networks (Rusu et al., 2016), Progressive Prompts (Razdaibiedina et al., 2023) extended this idea to apply PT in the TIL setting. At a time step $t$, we initialize a new trainable prompt $P_t$ which is then concatenated with all previously trained prompts $\{P_j\}_{j<t}$ and the text

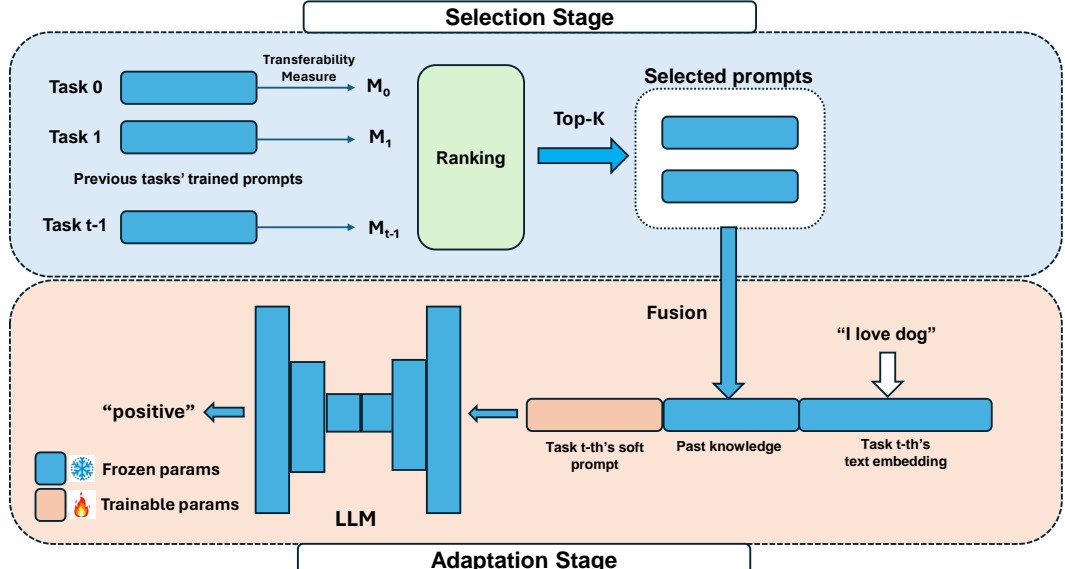

Figure 1: **Overview of our proposed framework:** In the selection stage, previous tasks' prompt is sorted by their transferability score, and top-$K$ prompts with the highest score is selected to construct the past knowledge, which is crucial for facilitating knowledge transfer of the new task adaptation. During the adaptation phase, we maintain the shared LM and past knowledge frozen and utilize PT to derive a distinct soft prompt for each task $t$.

embedding $X_t$ to create a new embedding $\hat{X}_t = [P_t, P_{t-1}, \ldots, P_1, X_t]$. For each task $T_t$, we obtain $P_t^*$ by maximizing the log-likelihood of $Y_t$ given $\hat{X}_t$ *i.e.*, $P_t^* = \arg\max_{P_t} \sum_{(x_t, y_t) \in T_t} \log p(y_t | \hat{X}_t)$. The prompt $P_t$ is trainable while prompts trained on previous tasks $P_{j<t}$ and the pre-trained language model weights $\Theta$ are frozen during task $t$ training. After training task $t$, we store $P_t^*$ for inference.

**Transferability measures**    In (Tran et al., 2019), *transferability* $\mathrm{Tr}(\mathcal{D}_s, \mathcal{D}_t)$ of source task $\mathcal{D}_s$ to target task $\mathcal{D}_t$ is defined as the expected log-likelihood on the training set of trained weights $w_s$ with $\mathcal{D}_s$ on target dataset $\mathcal{D}_t$ *i.e.*, $\mathrm{Tr}(\mathcal{D}_s, \mathcal{D}_T) = \mathbb{E}[\log p(w_s, \mathcal{D}_t)]$. However, an exact calculation of $\mathrm{Tr}(\mathcal{D}_s, \mathcal{D}_t)$ is not required to compare models in practice since we only need a measure having a positive correlation with $\mathrm{Tr}(\mathcal{D}_s, \mathcal{D}_t)$. Therefore, transferability measures (TM) are developed as an *computationally efficient* alternative for tasks or model comparison. In particular, a transferability measure $\mathcal{M}$ is a real-valued metric taking $(\mathcal{D}_s, \mathcal{D}_t)$ as inputs and returns a real value $\mathcal{M}(\mathcal{D}_s, \mathcal{D}_t) \in \mathbb{R}$. In cases where the source task $\mathcal{D}_s$ is unavailable, we can approximate $\mathcal{M}(\mathcal{D}_s, \mathcal{D}_t) \approx \mathcal{M}(w_s, \mathcal{D}_t)$. Given two pre-trained weights $w_1$ and $w_2$ learned from different sources, a target dataset $\mathcal{D}_t$, transferability scores $\mathcal{M}(w_1, \mathcal{D}_t)$ and $\mathcal{M}(w_2, \mathcal{D}_t)$, and the actual performance scores $\mathrm{Perf}(w_1, \mathcal{D}_t)$ and $\mathrm{Perf}(w_2, \mathcal{D}_t)$ of $w_1$ and $w_2$ on the target dataset $\mathcal{D}_t$, $\mathcal{M}$ should satisfy the following equation:

$$\mathrm{Perf}(w_1, \mathcal{D}_t) \leq \mathrm{Perf}(w_2, \mathcal{D}_t) \iff \mathcal{M}(w_1, \mathcal{D}_t) \leq \mathcal{M}(w_2, \mathcal{D}_t). \tag{1}$$

As indicated in Eq.(1), the pre-trained weight $w_2$ has a higher transferability score than $w_1$, and $w_2$ is expected to achieve higher performance on $\mathcal{D}_t$ than $w_1$.

Previous research (Bassignana et al., 2022) empirically shows that transferability estimators such as LogME (You et al., 2021) are more reliable than NLP practitioners in model selection for downstream adaptation. LogME employs the marginal likelihood $p(y|F)$ to measure the compatibility of target dataset features extracted by a source model $F \in R^{n \times d}$ with its corresponding labels $y \in R^n$. Here, $n$ is the size of the target dataset and $d$ is the feature dimension. Theoretically, $p(y|F) = \int_w p(y|F, w)p(w)dw$ is intractable since it requires integration over the space of $w$, where $w$ is the classification head on top of the extracted features. However, in (You et al., 2021), the authors provided an iterative approach based on optimization to estimate the density $p(y|F)$.

## 3 METHODOLOGY

### 3.1 IMPROVING FSCL VIA SELECTIVE KNOWLEDGE TRANSFER

Let $g$ be the base network initialized with pre-trained weights $\theta_0$. At a timestep $t$, we aim to learn a set of incremental model parameters $\theta_t$ for the current task $T_t$. We assume the existence of model parameters $\theta_t^{\text{past}}$ representing the past knowledge. The memory $\theta_t^{\text{past}}$ is employed jointly with $\theta_t$ to maximize the log-likelihood on the downstream task $T_t$ using gradient descent. Formally, we optimize the following objective function:

$$\theta_t^* = \arg\max_{\theta_t} \sum_{x,y \in T_t} \log p(y|x, \theta_t, \theta_t^{\text{past}}). \tag{2}$$

In the above equation, $\theta_t^{\text{past}}$ needs to be chosen wisely to ensure the maximum KT for task $t$. We establish $\theta_t^{\text{past}}$ from a subset of similar memories [1] $\mathcal{P}_K \subseteq \mathcal{P}$, where $\mathcal{P}$ is the set of all previous memories. Next, we introduce how to select $\mathcal{P}_K$, and then explain how to construct $\theta_t^{\text{past}}$ using $\mathcal{P}_K$.

**Selection** We introduce the selection process of past memories with the help of transferability measures. In *parameter isolation-based* CL approaches, a set of incremental parameters $\mathcal{P} = \{\theta_1, \ldots, \theta_{t-1}\}$ associated with previous tasks $\{T_1, \ldots, T_{t-1}\}$ are used to learn a subsequent task $T_t$. Note that the prior datasets $T_{j<t}$ are absent, and we only have the incremental parameters $\theta_{j<t}$ for measuring task similarity. We select the subset $\mathcal{P}_K \subseteq \mathcal{P}$ as the set of $K$ incremental parameters trained on the most prior similar tasks to the current task $T_t$, where $K$ is a hyper-parameter. To measure the similarity between a prior task $T_j$ and the current task $T_t$, we estimate the transferability score $s_j^t = \mathcal{M}(\theta_j, T_t)$ of the trained residual parameters $\theta_j$ on task $T_t$, noting that all feature-based transferability measures could be employed (You et al., 2021; Gholami et al., 2023).

**Aggregation** (Zhou et al., 2022) found that jointly training the current task with related ones outperforms training on all tasks and that key tasks should have higher weights during new task training. Inspired by this observation, we derive an aggregation mechanism called *weighted sum* where the past knowledge $\theta_t^{\text{past}}$ is calculated as a linear combination of all selected incremental weights $\theta_k \in \mathcal{P}_K$ with their corresponding transferability scores $s_k^t$ *w.r.t.* the current task as coefficients:

$$\theta_t^{\text{past}} = \frac{\sum_{k=1}^K s_k^t \theta_k}{\sum_{k=1}^K s_k^t}. \tag{3}$$

**Discussions** Our proposed framework is general, and can be applied to any parameter-efficient tuning approaches such as Prompt Tuning (PT) (Lester et al., 2021), Adapter (Houlsby et al., 2019) and LoRA (Hu et al., 2021). In Sec. 3.2, we devise *Log-evidence Progressive Prompts*, a representative of our proposed framework for prompt-based CL algorithms. Another advantage of our proposed framework is its efficiency, as it does not require training per task's representation (Zhang et al., 2015), which would otherwise add significant computational overhead due to the backward pass for gradient descent calculation. Finally, using TMs offers a useful interpretation of task similarity, *i.e.*, identifying which tasks should be jointly trained with the current task to maximize accuracy. The experiment in Sec. 4.4 demonstrates this capability of our proposed method.

### 3.2 LOG-EVIDENCE PROGRESSIVE PROMPTS (LePP)

We devise *Log-evidence Progressive Prompts (LePP)*, a two-stage CL algorithm based on Progressive Prompts using SKT. Given a set of trained prompts $\mathcal{P} = \{P_1, \ldots, P_{t-1}\}$ at a time step $t$, we propose learning a incremental soft prompt $P_t$ on a current task $T_t = (\mathcal{D}_t^{train}, \mathcal{D}_t^{test})$ following 2 steps: *prompt selection* and *prompt aggregation*. In the *prompt selection* step, we first extract the *encoder features* $F_t^j$ of the training dataset $\mathcal{D}_t^{train}$ for each trained prompt $P_j \in \mathcal{P}$. Then, the transferability score $s_t^j$ is calculated as the log evidence of the current task label $Y_t$ given the encoder features $F_t^j$, *i.e.* $s_j^t = \log p(Y_t|F_t^j)$. We select the top $K$ prompts with the highest scores $s_t$ as $\mathcal{P}_K$. In the *prompt*

---

[1] In this paper, we consider the incremental weights as past memories, and use the terms interchangeably.

---

**Algorithm 1** Log-evidence Progressive Prompts (LePP)

---

**Input:** Training sets $\mathcal{T} = \{T_1, T_2, \ldots, T_M\}$, $T_t = \{(x_t^i, y_t^i)\}_{i=1}^{N_t}$, a prompt pool $\mathcal{P} = \{\}$
1: **for** $t = 1, \ldots, M$ **do**
2:      Random initialize the $t$-th task's soft prompt $P_t$
3:      *# Selection stage for task $T_t$*
4:      **for** $P_j \in \mathcal{P}$ **do**
5:          Get the feature matrix $F_t^j = \text{Feat}(\mathcal{D}_t^{train}, P_j)$, $F_t^j \in \mathbb{R}^{N_t \times d}$
6:          Get the label vector $Y_t$
7:          Calculate transferability score $s_t^j = \log p(F_t^j, Y_t)$ using Alg.1 in You et al. (2021)
8:      **end for**
9:      **if** $|\mathcal{P}| \leq K$ **then**
10:          $\mathcal{P}_K \leftarrow \mathcal{P}$
11:      **else**
12:          $\mathcal{P}_K = \{P_1, \ldots, P_K\} \leftarrow K$ prompts with the highest transferability scores $s_t^k$ from $\mathcal{P}$
13:      **end if**
14:      *# Adaption stage for task $T_t$*
15:      Calculate $P_t^{past} = \frac{\sum_{k=1}^K s_t^k P_k}{\sum_{k=1}^K s_t^k}, \forall P^k \in \mathcal{P}_K$
16:      Optimize $P_t^* = \arg\max_{P_t} \sum_{(x_t, y_t) \in T_t} \log p(y|[P_t, P_t^{past}, X], \Theta)$.
17:      $\mathcal{P} = \mathcal{P} \cup \{P_t^*\}$
18: **end for**

---

*aggregation* step, the past prompt $P_t^{\text{past}}$ is the weighted combination of all selected prompts in $\mathcal{P}_K$. Formally, $P_t^{\text{past}} = \frac{\sum_{k=1}^K s_t^k P_k}{\sum_{k=1}^K s_t^k}$, where $s_t^k$ is the transferability score corresponding to $P_k$. Finally, we concatenate $[P_t, P_t^{\text{past}}, X]$ and utilize the same prompt tuning mechanism as in Sec. 2 to obtain the optimal prompt $P_t^*$ for the current task $T_t$. The detailed algorithm is provided in the Algorithm 1.

## 4 EXPERIMENTS

### 4.1 EXPERIMENT SETUPS

**Datasets** We evaluate our proposed method on two popular continual learning benchmarks for NLP. We first validate it on a short-stream few-shot CL learning benchmark introduced in (Qin & Joty, 2022). This benchmark contains four text classification datasets from (Zhang et al., 2015) including DBPedia (article classification), Amazon (sentiment analysis), Yahoo Answers (question answering), and AGNews (new classification). We randomly select 16 samples per class for training and hold out 500 samples per class for validation. We also consider the challenging long-stream text classification benchmark (Razdaibiedina et al., 2023), where the performance of CL algorithms is highly vulnerable to negative transfer. This benchmark contains 15 text classification tasks from different task types and domains. We randomly pick 10, 20, and 100 samples per class for training, while holding out 500 samples per class for validation. Task descriptions are provided in the Appendix.

**Baselines** We compare our proposed method with traditional CL baselines and prompt-based CL baselines for NLP tasks. In general, sequential fine-tuning (FT), prompt tuning (PT), and experience replay (ER) are employed. For T5-based models, we include Progressive Prompts (Razdaibiedina et al., 2023) and LFPT5 (Qin & Joty, 2022). For BERT-based models, we compare our proposed method with IDBR (Huang et al., 2021), MBPA++ (de Masson D'Autume et al., 2019), and Progressive Prompts (Razdaibiedina et al., 2023).

**Metrics** We calculate the Average Accuracy (AA) over $M$ tasks to measure the effectiveness of our proposed method. Formally, after training on task $M$, AA is calculated as $AA_M = \frac{1}{M} \sum_{m=1}^M \text{Acc}(m, M)$, where $\text{Acc}(t, M)$ is the accuracy on the test set of $t$-th task after the last

Table 1: **Average performance on long-sequence experiments of the proposed algorithm compared to baselines using the BERT-based model.** All results are averaged over 5 runs. Asterisk indicates models trained the entire models while others only train a soft prompt. **Bold** indicates the best results. Our proposed method outperforms baselines by a wide margin.

| Method ↓ | Order8 | | | Order9 | | | Order10 | | |
|---|---|---|---|---|---|---|---|---|---|
| Num samples → | 10 | 20 | 100 | 10 | 20 | 100 | 10 | 20 | 100 |
| FT* | 38.44 | 29.92 | 33.94 | 37.74 | 30.52 | 38.64 | 34.95 | 33.65 | 40.32 |
| PT | 39.58 | 34.98 | 54.53 | 30.6 | 39.31 | 54.83 | 33.49 | 34.91 | 34.77 |
| ER* | 31.97 | 50.67 | 35.37 | 32.81 | 50.68 | 36.93 | 38.49 | 50.63 | 39.62 |
| IDBR* (2021) | 33.94 | 39.73 | 38.21 | 39.45 | 37.96 | 40.81 | 34.07 | 32.90 | 36.74 |
| Progressive (2023) | 54.03 | 55.59 | 61.66 | 53.89 | 56.68 | 58.31 | 54.00 | 54.12 | 64.65 |
| LePP(ours) | **56.12** | **58.91** | **64.01** | **56.96** | **57.77** | **63.61** | **56.35** | **58.48** | **65.28** |

task training is completed. In other words, the average accuracy demonstrates the system's average performance after training on the final task.

## 4.2 IMPLEMENTATION DETAILS

**Backbones** SKT is a model-agnostic method for CL that can be integrated into any backbone. For NLP, we employ three main backbones with different scales and architectures including $\mathbf{BERT}_{base}$ (Devlin et al., 2018) with **110M** parameters, and T5 backbones ($\mathbf{T5}_{small}$ and $\mathbf{T5}_{large}$ with **60M** and **770M** parameters, respectively). We use the model implementations and pre-trained weights from Hugging Face (Wolf et al., 2020) to ensure consistent results.

**Training configurations** We conducted experiments with $\mathbf{T5}_{small}$ and $\mathbf{BERT}_{base}$ on 4 NVIDIA A5000 GPUs, while experiments with $\mathbf{T5}_{large}$ were conducted on 2 NVIDIA A40 GPUs. We used hyperparameters as indicated in (Razdaibiedina et al., 2023) for a fair comparison. In particular, we employ Adam optimizer with a learning rate of 0.3 and a batch size of 8. We set the prefix length of the soft prompt to 10 tokens per task. Since T5 models are encoder-decoder models, we utilize the same text-to-text format as in Progressive Prompts for training T5 models. For example, the label "0/1" is converted to "positive/negative" for text generation. For BERT-based models, the prefix length is 50 tokens per task and the learning rate is chosen at 0.1. In addition, we applied the reparameterization trick via a two-layer MLP with 800 hidden neurons in each layer, which can be discarded during inference. Since the BERT-based model is encoder-only, we keep the original task labels and train an additional linear layer on top of the encoder to classify sentences. In our experiments, we selected five prompts with the highest transferability scores for knowledge aggregation.

## 4.3 MAIN RESULTS

**Results on the long stream benchmark** We first validate our framework with the $\text{BERT}_{base}$ model, one of the most popular language models based on the transformer architecture. We ran experiments with three different orders as in (Razdaibiedina et al., 2023), their details are in the Appendix, and report results in Tab. 1. *LePP* consistently outperforms other baselines across all three orders with a large margin. Interestingly, even with only 10 samples per class, *LePP* yields significant improvements, achieving 2.09% to 3.07% higher average accuracy than the second-best method. *LePP* is a model-agnostic method that can work with any backbone, so we employ T5-based models in our experiments. Due to the limited computational resources, we ran experiments five times with a random task order and reported the average results obtained by training $\mathbf{T5}_{small}$ and $\mathbf{T5}_{large}$ on different dataset sizes with 10, 20, and 100 samples per class, as shown in Tab. 2. The results indicate that LePP outperforms other baselines and consistently yields substantial improvements over the previous state-of-the-art, Progressive Prompts. Specifically, for $\mathbf{T5}_{small}$, our approach achieves up to 4.46%, 3.2%, and 3.15% higher average accuracy than Progressive Prompts when training with 10, 20, and 100 samples per class, respectively. For $\mathbf{T5}_{large}$, despite performance saturation due to the number of model parameters, our method still provides additional boosts in average accuracy. For instance, LePP's accuracy is up to 1.73 % higher than that of Progressive Prompts. These results

Table 2: **Average performance on long-sequence experiments of the proposed algorithm compared to baselines using T5 models.** All results are averaged over 5 runs. Asterisk indicates models trained the entire models while others only train a soft prompt. **Bold** indicates the best results. Our proposed method outperforms baselines.

| Method ↓ | Random order | | | Method ↓ | Random order | | |
|---|---|---|---|---|---|---|---|
| Num samples → | 10 | 20 | 100 | Num samples → | 10 | 20 | 100 |
| **T5-small** | | | | **T5-large** | | | |
| MTL* | 60.93 | 66.93 | 61.86 | MTL* | 39.73 | 70.72 | 70.44 |
| FT* | 38.73 | 40.36 | 43.63 | FT* | 41.82 | 41.73 | 40.29 |
| ER* | 55.89 | 59.37 | 61.01 | ER* | 49.39 | 53.91 | 34.66 |
| LFPT5 (2022) | 25.64 | 31.84 | 32.05 | LFPT5 (2022) | 36.12 | 32.94 | 42.94 |
| Progressive (2023) | 59.11 | 63.86 | 66.62 | Progressive (2023) | 74.05 | 77.24 | 78.93 |
| LePP (Ours) | **63.57** | **67.38** | **69.77** | LePP (Ours) | **75.05** | **77.55** | **80.66** |

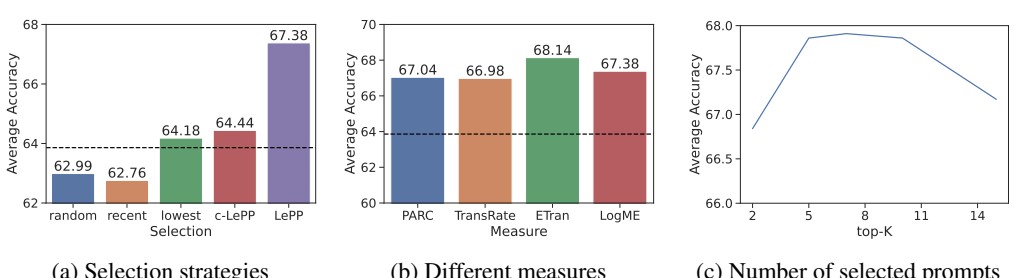

(a) Selection strategies     (b) Different measures     (c) Number of selected prompts

Figure 3: **Ablation studies on hyper-parameter choices**. The dashing line indicates the Progressive Prompts' performance. (a) Selecting prompts with the highest scores yields the best result. (b) Re-weighting prompts with TMs consistently outperforms Progressive Prompts and more accurate TMs lead to higher AAs. (c) Using a proper number of prompts yields significant improvements.

demonstrate the robustness of our proposed method which can effectively select useful information for learning new tasks and remains scalable with the size of language models and architecture.

**Results on the standard continual learning benchmark** We also conduct experiments on a standard few-shot continual learning benchmark with $\text{T5}_{large}$ (Qin & Joty, 2022) to illustrate the effectiveness of our proposed method with short sequences and its robustness to task order. We run experiments with five random task orders and 16 samples per class, and report the average results in Fig. 2. Notably, LePP surpasses other baselines on the short stream benchmark and boosts Progressive Prompts's performance. Particularly, LePP consistently outperforms Progressive Prompts, achieving 1.7% higher average accuracy. This demonstrates that our method can autonomously select relevant knowledge regardless of task orders, thereby improving the overall system performance.

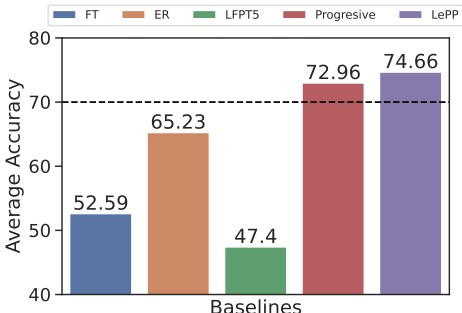

Figure 2: Average accuracy on standard continual learning benchmark of **LePP** compared to baselines.

## 4.4 ABLATION STUDIES

**Task selection and knowledge aggregation** In our experiments, we select the five most transferable tasks for new task adaptation and aggregate selected prompts by weighted averaging. To validate

Table 3: **Task selection frequency of each target dataset in the long-sequence benchmark.**

| task | Selected prior tasks |
| --- | --- |
| qqp | copa(3), wic(3), multirc(3), cb(3), boolq(3) |
| rte | cb(3), mnli(2), wic(2), boolq(2), qqp(2), multirc(2), copa(2) |
| imdb | boolq(3), wic(3), qqp(3), copa(3) |
| sst2 | copa(3), boolq(3), imdb(3), cb(2) |
| dbpedia | boolq(3), cb(3), multirc(2), copa(2) |
| ag | imdb(2), qqp(2), multirc(2), boolq(2), cb(2), copa(2) |
| yelp | imdb(2), ag (2), copa(3), boolq(2), cb(2), wic(2) |
| amazon | ag(2), yelp(2), qqp(2), boolq(2), cb (2), copa(2) |
| yahoo | ag(3), imdb(2), yelp(2), boolq(2), cb(2), copa(2) |

this design choice, we conduct an ablation study comparing it with other selection methods: (1) random selection, (2) 5 most recent tasks, (3) 5 least transferable tasks, and naively concatenate all the selected prompts as in Progressive Prompts (Razdaibiedina et al., 2023). We run experiments on the long-sequence benchmark with the $\mathbf{T5}_{small}$ backbone, using the same settings as in Sec. 4.2, and report results in the Fig. 3a. As expected, selecting the most transferable tasks yields the greatest improvement over other baselines, while learning from random sources can disrupt the learning process of the target task. Additionally, learning from the most recent tasks can decrease accuracy if those tasks are irrelevant to the current one. Interestingly, learning from the least transferable tasks slightly improved overall performance, with an increase of 0.32% over Progressive Prompts. Our results support previous findings Paredes et al. (2012) that learning from unrelated tasks can be beneficial in MTL. Since our weighted ensemble learning procedure partially resembles the MTL mechanism, therefore it can inherit this property. However, selecting the most relevant tasks remains the preferred strategy with the highest average accuracy. We additionally compare the knowledge aggregation mechanisms including selected prompts concatenation *c-LePP* and weighted averaging *LePP*. Fig. 3a shows that weighted averaging outperforms concatenation by a significant margin of 2.93%, demonstrating that transferability scores provide useful information for learning a new task using previous knowledge.

**Different transferability measures** In LePP, we employ an evidence-based metric to estimate the task transferability, therefore, it is also compatible with other transferability estimation metrics. In this ablation study, we compare LogME with other metrics, including PARC (Bolya et al., 2021), TransRate (Huang et al., 2022), and ETran (Gholami et al., 2023). We replicate the experiment with $\mathbf{T5}_{small}$ on the long task sequence benchmark with the same hyperparameters as indicated in Sec. 4.2 and report the results in Tab. 3b. As shown in the table, using transferability measures to weight the task importances yields substantial improvements over the Progressive Prompts' performance, therefore it is robust to the TM choice. Notably, ETran has the highest average accuracy among TMs since it takes into account the fact that a target dataset could be in-distribution regarding a source model. Therefore, it can further eliminate noisy prior source models with respect to future tasks.

**The number of selected tasks** $K$ We examine how the number of selected tasks affects our final results. We conduct experiments with $\mathbf{T5}_{large}$ on the long-sequence benchmark as indicated in Sec. 4.2. We select the 2, 5, 7, and 10 highest-scoring prompts to compare them with using all prompts. The selected prompts are aggregated in a *weighted sum* manner. The experiment results are reported in Fig. 3c. Fig. 3c shows that selecting only the two most relevant prompts already provides benefits over Progressive Prompts. As the number of selected tasks increases, more useful information can be discovered from the task sequence. However, the average accuracies remain relatively stable when selecting between 5 to 10 prompts. Interestingly, the system's performance degrades substantially when we utilize all previous prompts. This indicates that not all previous prompts are informative for future tasks, and discarding irrelevant prompts leads to further enhancements.

**Analyses on task similarity** We investigate the task-selected frequency from three different runs during continual training of the Bert-base-uncased model with a long sequence (order 9) using 100 samples per class. Table 3 shows the selected tasks and their frequency appearing in the top five selected tasks when training on the current dataset. The results confirm that LogME can autonomously

identify similar tasks from the sequence, and prioritize them in the top-5 selection. For example, CB and MNLI, natural language inference tasks, were chosen for learning RTE, another nature language inference task. A similar phenomenon is observed for sentiment analysis tasks such as Yelp-IMDB and Amazon-Yelp. In addition, QA tasks (BoolQ, Multirc) stand out as the most frequently selected task type selected by LogME. One possible explanation for this observation is that the QA tasks involve high-level reasoning which could benefit other tasks, and QA tasks are often used as source tasks for pre-training (Jia et al., 2021). We observed the same finding as (Wu et al., 2024) that CB as a source task yields positive transfer for many target tasks. Interestingly, our metric can detect dissimilar source-target pairs but could result in positive transfers such as WIC-MultiRC, WIC-Yahoo, CB-QQP. This interpretation is useful if more data belonging to a task or related tasks becomes available in the future, as we would not have to re-estimate the task similarity for training this task or training a sequence from scratch.

**Results on computer vision benchmarks** Our proposed framework is general and can be applied to other modalities. To validate its generalization, we conduct experiments on two popular large-scale computer vision datasets, Split-CIFAR-100 and Split-CUB-200. We randomly split each dataset into 10 tasks, with 10 and 20 classes for CIFAR-100 and CUB-200, respectively. We incorporate SKT into the ensemble step of HiDE-Prompt to derive *SKT-HiDE*. Specifically, all selected previous prompts are weighted by their transferability

Table 4: Average accuracy on two computer vision benchmarks Split-CIFAR-100 and Split-CUB. Higher is better. Incorporating our proposed strategy increases the performance of HiDE-Prompt.

| Method | Split-CIFAR | Split-CUB |
|---|---|---|
| L2P (2022c) | 97.64 | 79.09 |
| DualPrompt (2022b) | 97.74 | 80.18 |
| SPrompt (2022a) | 97.46 | 78.36 |
| HiDE-Prompt (2024a) | 97.72 | 82.22 |
| SKT-HiDE (Ours) | **98.13** | **82.90** |

score $p_t = \alpha \sum_{k=1}^{K} s_k^t e_k + (1-\alpha)e_t$, where $s_k^t$ is the transferability score of task $k$ prompt on the current task $t$. We select the three most transferable prompts for learning the new task $t$, and use the same hyperparameters as in Wang et al. (2024a). Averaged results from 3 runs are reported in Tab. 4. We can observe that selecting highly transferable prompts for learning a new task consistently yields additional improvements with 0.41% and 0.68% over HiDE-Prompt, the current state-of-the-art algorithm for computer vision CL, on Split-CIFAR-100 and Split-CUB-200, respectively. This demonstrates the generalization of our proposed framework, which works effectively across different modalities including text and images.

## 5 RELATED WORKS

**Continual learning** In continual learning (CL), neural networks are prone to catastrophic forgetting when trained sequentially on a sequence of tasks with non-stationary data. To address this, constraints can be placed on the parameter space to maintain stability, such as Elastic Weight Consolidation (EWC) (Kirkpatrick et al., 2017) adding regularizers to keep task parameters close, Progressive Networks (Rusu et al., 2016) freezing trained parameters, and Dynamically Expandable Network (Yoon et al., 2018) identifying key parameters for new tasks via retraining. In language models, these ideas enhance robustness to forgetting and knowledge sharing between tasks. For instance, IDBR (Huang et al., 2021) disentangles task-specific and task-generic information, while CTR (Ke et al., 2021) and CAT (Ke et al., 2021) introduce CL plug-in modules between BERT layers for knowledge sharing, but these methods face inefficiencies with large models due to whole network training, task masking, and memory constraints. Especially, those methods are only tested on either NLP tasks (Ke et al., 2020; Huang et al., 2021; Ke et al., 2021; Razdaibiedina et al., 2023) or CV tasks (Nguyen et al., 2017; Yoon et al., 2018; Wang et al., 2022b;c; 2024a).

**Parameter-efficient Fine-tuning** Parameter-efficient Fine-tuning (PEFT) (Houlsby et al., 2019; Liu et al., 2022; Li & Liang, 2021; Hu et al., 2021) has demonstrated significant success in fine-tuning foundational language models with a remarkable reduction in parameters. Building on this success in single-task adaptation, several approaches (Madotto et al., 2021; Peng et al., 2024; Zhang et al., 2024b) have been devised to extend its application to continual learning (CL) settings. For example,

AdapterCL (Madotto et al., 2021) trains a separate adapter for each task to facilitate the continual training of task-oriented dialog systems. LFPT5 (Qin & Joty, 2022) addresses the few-shot language learning problem by learning a large soft prompt shared among tasks to enhance knowledge sharing. PTCC (Zhang et al., 2024b) recalibrates the weights of each trained soft prompt for initializing new tasks based on the similarity between tasks in both context and label spaces. However, these approaches suffer from several limitations, including limited knowledge sharing (Madotto et al., 2021), catastrophic forgetting (Qin & Joty, 2022), task interference (Razdaibiedina et al., 2023), and limited scalability with the number of tasks (Peng et al., 2024; Zhang et al., 2024b).

**Transferability measures** Selecting the pre-trained model that can achieve the highest performance after fine-tuning poses a significant challenge, given the vast number of off-the-shelf pre-trained models available on model hubs (Wolf et al., 2020; maintainers & contributors, 2016), and the impracticality of brute-force fine-tuning. Transferability measures (Nguyen et al., 2020; 2023; Huang et al., 2022; Tran et al., 2019; You et al., 2021; Zhang et al., 2024a), which produce a real-value score correlated with the actual model performance for model ranking, have emerged as a promising solution to this problem. Although transferability measures have shown success in computer vision tasks (Agostinelli et al., 2022; Wang et al., 2024b), their application in NLP tasks is still limited to single-task transfer learning (Bassignana et al., 2022), while their potential in continual learning (CL) remains largely unexplored. In a recent work (Bassignana et al., 2022), authors highlight that LogME provides a more robust method for ranking model performance compared to the intuitions of NLP practitioners. Motivated by this observation, we take the first step in applying transferability measures to facilitate the forward transfer of CL systems.

# 6 CONCLUSIONS AND LIMITATIONS

**Conclusions** We present *selective knowledge transfer* (SKT), a novel framework for few-shot continual learning with LMs, to advance the positive forward transfer for learning future tasks. Our proposed framework is computationally efficient and can autonomously reveal similar and dissimilar tasks, therefore its scalability is guaranteed with the length of the task stream. We develop Log-evidence Progressive Prompts *LePP*, an enhanced version of Progressive Prompts following our proposed principle. Extensive experiments confirm our proposed framework can significantly leverage existing SoTAs for continual learning with NLP and CV.

**Limitations** This study has several limitations. First, it focuses solely on classification tasks, excluding other NLP tasks like text generation. This exclusion arises because existing transferability measures mainly cater to classification and regression tasks. Although text generation could be treated as a regression task, this approach neglects the intricate relationships between words in sentences, leaving the challenge of developing an efficient transferability measure for generative models unresolved. Second, our framework relies on a hyper-parameter K to select the optimal amount of relevant knowledge, which is a sub-optimal ideal. A potential solution is to use a greedy algorithm for a near-optimal selection or develop a measure to identify tasks that might cause negative transfer. These aspects remain open for future exploration.

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

Table 5: Descriptions of 15 datasets in long sequence CL experiments.

| Dataset | Collection | Type | Domain |
|---|---|---|---|
| yelp | CL benchmark | sentiment analysis | Yelp reviews |
| amazon | CL benchmark | sentiment analysis | Amazon reviews |
| dbpedia | CL benchmark | topic classification | Wikipedia |
| yahoo | CL benchmark | QA | Yahoo Q&A |
| ag | CL benchmark | topic classification | news |
| mnli | GLUE | NLI | various |
| qqp | GLUE | paraphrase detection | Quora |
| rte | GLUE | NLI | news, Wikipedia |
| sst2 | GLUE | sentiment analysis | movie reviews |
| wic | SuperGLUE | word sense disambiguation | lexical databases |
| cb | SuperGLUE | NLI | various |
| copa | SuperGLUE | QA | blogs, encyclopedia |
| boolq | SuperGLUE | boolean QA | Wikipedia |
| multirc | SuperGLUE | QA | various |
| imdb | other | sentiment analysis | movie reviews |

Table 6: Task orders for training BERT-based models.

| Task | Order |
|---|---|
| Order 8 | mnli, cb, wic, copa, qqp, boolq, rte, imdb, yelp, amazon, sst2, dbpedia, ag, multirc, yahoo |
| Order 9 | multirc, boolq, wic, mnli, cb, copa, qqp, rte, imdb, sst2, dbpedia, ag, yelp, amazon, yahoo |
| Order 10 | yelp, amazon, mnli, cb, copa, qqp, rte, imdb, sst2, dbpedia, ag, yahoo, multirc, boolq, wic |

# A  APPENDIX

## A.1  TASK DESCRIPTION

Table. 5 contains the details of 15 datasets used in our long sequence experiments. These datasets are collected from different NLP benchmarks including CL benchmark (Zhang et al., 2015), GLUE (Wang et al., 2018), and SuperGLUE (Wang et al., 2019). They encompass a variety of tasks *e.g.*, sentiment analysis, topic classification, question answering, and natural language inference (NLI). Additionally, the text data comes from a wide range of domains including movie reviews, Amazon reviews, Wikipedia, and so on. As a result, learning under these settings is susceptible to negative transfer, highlighting the need for accurate task selection.

## A.2  TASK ORDER

We provide detailed task orders to train BERT-based models in Tab. 6 to investigate the robustness of our proposed method with task order.

## A.3  TRAINING

As in (Razdaibiedina et al., 2023), we use a batch size of 8 for all experiments except for multitask training, where the batch size is set to 2 due to the VRAM limitations of GPUs. We also vary the number of training epochs depending on the number of samples. Specifically, for 10 and 20 samples per class, we train our model for 300 epochs for each task. For 100 samples per class, the model is trained for 150 epochs. In terms of random initialization, we use the same initialization technique as Lester et al. (2021), where new prompts are initialized by randomly sampling tokens in the embedding layer.

---

**Algorithm 2** Feature extraction (Feat)

---

**Input:** Dataset $\mathcal{D} = \{x_i, y_i\}_{i=1}^N$, a trained prompt $P$, encoder model $\text{Enc} = (\text{emb}, h)$
**Output:** Feature matrix $F$
1: **for** $i = 1, \ldots, N$ **do**
2:     Get the embedding vector $X_i = \text{emb}(x_i)$ of $x_i$
3:     $f_i = h([P; X_i])$
4: **end for**
5: $F = [f_1, f_2, ..., f_N]^T$

---

Table 7: **Total training time comparison on Split-CIFAR.** *SKT-HiDE* yields better performance while occur a minimal computational overhead.

| Method | Running Time |
|---|---|
| HiDE-Prompt (2024a) | 2:07:43s |
| *SKT-HiDE* (ours) | 2:08:35s |

## A.4 ALGORITHM

This section outlines the procedure for extracting features on a target dataset $\mathcal{D}$ using a trained prompt $P$. Firstly, for each sample, we feed the tokenized input $x_i$ through the embedding layer to obtain the embedding vector $X_i$. Then, the trained prompt $P$ is prepended to embedding $X_i$, resulting in the prompted embedding$[P; X]$, which is then forwarded to h, the rest of the encoder model, to output the feature $f_i$. This process is repeated for every sample in the dataset $\mathcal{D}$ to get the feature matrix $F$.

## A.5 RUNNING TIME

We report the total running time on Split-CIFAR in Tab. 7. *SKT* can combine with other prompt-based CL algorithms to yield additional improvements with a minimal computational overhead.

## A.6 PER-TASK PERFORMANCE

We report the per-task accuracies to demonstrate the effectiveness of our proposed method. In particular, Tab. 8 contains the averaged accuracies of BERT models with order-8 using 100 samples per-class while in Tab. 9 reports the averaged accuracies of T5-small model in a single order with 20 samples. Our approach consistently improves each task performance, therefore increases the system performance overall.

Table 8: **Accuracy of each task in a long sequence using order-8 and BERT-based model**. All results are averaged over 5 runs. Clearly, *LePP* consistently increases accuracies of 9/15 tasks, and outperforms Progressive Prompts.

| Task | Progressive | LePP (Ours) |
|---|---|---|
| mnli | 45.62 | 45.69 |
| cb | 71.42 | 71.43 |
| wic | 51.59 | **53.92** |
| copa | 48.67 | **51.33** |
| qqp | 69.80 | **72.38** |
| boolq | **53.00** | 52.70 |
| rte | **54.43** | 51.07 |
| imdb | 69.93 | **75.10** |
| yelp | 47.43 | **48.95** |
| amazon | 38.49 | **47.85** |
| sst2 | 75.15 | **84.10** |
| dbpedia | 98.43 | **98.66** |
| ag | 86.92 | **87.20** |
| multirc | **49.43** | 49.03 |
| yahoo | 70.80 | 70.76 |
| average | 61.66 | **64.01** |

Table 9: **Accuracy of each task in a long sequence using T5-small**. **Bold** indicates the best results. Clearly, *LePP* consistently increases accuracies of 9/15 tasks, and outperforms Progressive Prompts.

| Task | Progressive | LePP (Ours) |
|---|---|---|
| boolq | 51.92 | 52.60 |
| imdb | 86.24 | 87.35 |
| rte | **58.99** | 53.59 |
| sst2 | **89.05** | 87.06 |
| dbpedia14 | 92.50 | 92.52 |
| cb | 77.14 | **77.68** |
| copa | 46.40 | **46.50** |
| amazon | **34.98** | 34.00 |
| wic | 54.10 | **55.49** |
| yelp | 42.53 | **43.72** |
| yahoo | 58.03 | **63.06** |
| qqp | 74.24 | **87.85** |
| ag | 77.84 | **79.85** |
| multirc | 56.12 | **73.65** |
| mnli | 57.84 | **70.96** |
| average | 63.86 | **67.06** |

