# OpenReview forum: "Advancing Few-shot Continual Learning via Selective Knowledge Transfer"
_ICLR.cc/2025/Conference — Submitted to ICLR 2025_

### Official Review · Reviewer_RQpU · 2024-10-27

**Soundness:** 2
**Presentation:** 2
**Contribution:** 2
**Rating:** 5
**Confidence:** 4

**Summary:**

This paper introduces Selective Knowledge Transfer (SKT) for few-shot continual learning with large language models (LLMs), aiming to enhance positive forward transfer when learning new tasks. The proposed framework is computationally efficient and can automatically identify both similar and dissimilar tasks, ensuring scalability as the task stream grows. The authors also present Log-evidence Progressive Prompts (LePP), an improved version of Progressive Prompts. In this approach, prompts from previous tasks are selected based on transferability scores, and their weighted sum is used as additional input for the LLMs. Extensive experiments demonstrate that the proposed framework significantly improves upon state-of-the-art methods for continual learning across both natural language processing and computer vision tasks.

**Strengths:**

* The proposed method is simple and effective.

* Experiments are extensive.

**Weaknesses:**

* The proposed method has limited novelty since it is a simple variant of the Progressive Prompts, proposed in [1] with prompt selection. Specifically, the proposed method replaces the prompt concatenation with prompt weighted average. Furthermore, weighted knowledge transfer has been widely studied in various fields. For example, in domain adaptation [3], continual learning [4], although the methods are slightly different, the principle is about the same.


* The proposed transferability scores is also proposed in [2], thus the metirc is not new in this paper. Also, the authors state that the previous task data is not available when learning new task, so they use the previous prompt parameters applied on new task to calculate the transferability metric. There is no analysis whether this way of calculation is suitable or not.

* Lack of deeper analysis on how the proposed approach with prompt selection improves the continual learning performance.  For example, select which part of prompt knowledge is helpful for learning new task.



Reference:

[1] Progressive prompts: Continual learning for language models, ICLR 2023

[2] Logme: Practical assessment of pre-trained models for transfer learning, ICML 2021

[3] Multi-source domain adaptation via weighted joint distributions optimal transport, ICML 2022

[4] Weighted Ensemble Models Are Strong Continual Learners, ECCV 2024

**Questions:**

N/A

---

> ### Author Response · Authors · 2024-11-26
>
> Weaknesses
>
> W1: We strongly disagree with the reviewer regarding our method being a simple variant of the Progressive Prompt. Our work introduces a general framework with the prompt selection mechanism to leverage the forward transfer between tasks. The derivation of LePP is a special case of our proposed framework, we also integrate our proposed framework to HiDE-Prompt to introduce a new algorithm. Our work replaced the prompt concatenation to make the proposed method general since concatenating too many prompts can deteriorate new task learning in the prompt-based methods (the total length of the prior prompts exceeds the context length of the model). Despite the simplicity of the weighted ensemble, the novelty of our work is in how we efficiently obtain the weights for knowledge aggregation which lies in the application of transferability scores, which is underexplored in the NLP domains.
>
> W2: Could you please provide more details about measuring the comparability of “this way of calculation”? Since we do not have the right to access the training data of the previous tasks, we only have the task prompts for comparability measurement. In addition, in [2], transferability scores are aimed to correlate with the model performance therefore it could correlate with the importance of weights up to a constant. When we aggregate in the weight manner, the constant is canceled and the correlation is preserved.
>
> W3: We analyzed the selected prior tasks during a new task training in Section 4.4 in the main paper. Our analysis agreed with many previous findings as indicated in the literature. We also provide the performance on each task here to demonstrate how prompt selection improves the CL performance. In general, prompt selection optimizes per-task performance and increases the system performance overall, please see Appendix A.6 for per-task performance.

---

> > ### Comment · Reviewer_RQpU · 2024-11-26
> > **Thanks**
> >
> > Thanks for your rebuttal. The rebuttal have addressed part of the concerns. I increased my score.

---

### Official Review · Reviewer_AZi7 · 2024-10-29

**Soundness:** 2
**Presentation:** 2
**Contribution:** 2
**Rating:** 5
**Confidence:** 4

**Summary:**

This paper explores few-shot continual learning with large language models and proposes a new strategy called selective knowledge transfer, which aims to identify and leverage relevant information from past knowledge to enhance performance. Additionally, it introduces Log-evidence Progressive Prompts (LePP), a two-stage continual learning algorithm that incrementally learns a soft prompt . It selects the top K trained prompts based on their transferability scores derived from encoder features.

**Strengths:**

1. Complete and clear algorithmic flow
2. This method is lightweight and efficient, , incurring minimal additional costs.
3. The experiments yield exceptional results.

**Weaknesses:**

1. All experiments are limited to classification with small datasets, which may restrict the paper's generalizability and contributions to other applications.

2. "However, more accurate measures have higher accuracies. " This sentence is a bit unclear.

3. Also, why was LogME chosen when ETran demonstrates better performance?

4. One of the key proposals of this paper is the introduction of transferability measures into continual learning; however, it lacks sufficient evidence and experiments to support its advancements and to clearly differentiate these measures from standard similarity metrics, such as cosine similarity. At present, it is difficult to see its superiority and necessity.

**Questions:**

1. A slight suggestion:  there are too many abbreviations in this paper. While some contribute to conciseness, others may cause confusion.
2. "Learning from the least transferable tasks slightly improved overall performance..." This conclusion is unclear and appears contradictory. The explanation is not fully convincing, as 'unrelated tasks' are not necessarily equivalent to 'harder tasks.'

---

> ### Author Response · Authors · 2024-11-26
>
> Weaknesses
>
> W1: We disagree with the reviewer in terms of the generalisability and contributions of our work. In the text classification problems, many datasets could be framed as regression tasks such as “yahoo_answer_topics” and the LogME can estimate the transferability between regression tasks. For instance, given a specific length for a text generation task, we can reformulate it as a high-dimension regression problem and LogME could be applied. For the “small” dataset, we conducted experiments on two popular vision-CL datasets: Split-CIFAR and Split-cub-200 datasets, which contain 5000 and 1200 images, respectively. In addition, we empirically show that our proposed method can work with both text and image datasets. Therefore, our proposed method can be generalised in various settings and applications.
>
> W2: We made a change to make this sentence clearer. Please refer to the lines 351-352 in the main paper.
>
> W3: We chose LogME in the main paper due to its simplicity for hyper-parameter tuning. For the ETran, to obtain optimal results, the coefficients between the energy term and the transferability term are required to be chosen carefully for better results, which is not suitable for the ultimate goal of the paper: an efficient framework with minimal hyper-parameter tuning. However, developing a measure tailored for LMs is still an open question and out of the scope of this work.
>
>
> W4: We emphasize that our proposed method is based on a critical assumption that we do not have the right to probe on the new task. This assumption is essential in settings where fast adaptation is required such as Online CL. Unlike our proposed method, they need a prompt trained on the current task to evaluate the similarity with prior prompts via the cosine metric. Obtaining the “probe” prompt is also problematic as the prompt’s performance depends on hyperparameter tuning. In addition, probe training incurs significant computational overhead since it requires both forward and backward passes, our proposed method only requires a single forward pass.
>
> Questions
>
> Q1. We thank the reviewer for your feedback. We will update the abbreviations for better understanding.
>
> Q2. We thank the reviewer for helping us improve the manuscript. Based on your recommendation, we updated the explanation to clarify your concerns. Please line the 400-402 in the newest version.

---

> > ### Comment · Reviewer_AZi7 · 2024-11-28
> >
> > Hi,
> >
> > Thank you for your response. I have reviewed the comments and still believe that this work has significant potential for improvement. Therefore, I will maintain my score for now. I hope to see further enhancements in the future.
> >
> > Best regards,

---

### Official Review · Reviewer_FhGR · 2024-11-03

**Soundness:** 2
**Presentation:** 2
**Contribution:** 2
**Rating:** 5
**Confidence:** 4

**Summary:**

This paper proposed selective knowledge transfer, a continual learning framework for LLMs utilizing transferability measures. It can be integrated into the prompt-based continual learning frameworks, leveraging related past knowledge to improve system performance.

**Strengths:**

It is a good point to consider knowledge transfer in continual learning, and combining it with prompt tuning is reasonable and effective.

**Weaknesses:**

1. The proposed method LePP heavily relies on previous work LogME and Progressive Promt. It is applying the LogME to Progressive Promt within the already established research question. As the research question is established with benchmarks, and the method is an A+B, this work only provides a limited technical contribution.
2. The author claims selective knowledge transfer is a novel and principled framework for continual learning with **LLMs**. However, the current knowledge transferability calculation requires the model to have the ability to encode features. As discussed by the authors in the limitation, LePP cannot be directly applied to decoder-only models which are the majority of "LLMs". As a reference, T5 is not claimed as "LLM" in Progressive Promt. This is an overclaiming. The influence of this work is limited by the supported task type and model type.
3. The writing could be further improved. Will list the writing problems in the questions section.

**Questions:**

1. Line 15~16, two points to improve with "This work presents selective knowledge transfer (SKT), a novel and **principled** framework towards continual learning with **LLMs**." First, I would suggest the authors carefully reconsider using "principled" as a self-comment. It is always better to provide a good presentation to let readers feel the method is principled, not commented by the authors themselves. Second, "LLMs" is somehow overclaimed as T5 is not always considered a LLM.
2. Line 32, "Modern language models must efficiently adapt to dynamic environments", "must"--> "are expected to" / "are required to"/ "need to", whatever appropriate word instead of must.
 Another problem with this sentence is that decoder-only modern LLMs can adapt to dynamic environments with in-context learning.
3. Line 36~39, "However, two main obstacles ...  catastrophic forgetting (CF),... and forward transfer (FT), ..." The word "obstacles" needs consideration. Should forward transfer be considered an obstacle or a research question/methodology?
4. Table 1 caption is too short and void. Should put more information into the caption to make the table self-contained. The author can refer to the Progressive Prompt paper.
5. Line 504, the authors claim their proposed framework is computational-efficient, "Our proposed framework are computational-efficient...".  "are"-->"is", and although theoretically, this may be true, the authors should provide perceptible results to demonstrate how
computational-efficient their framework is.

---

> ### Author Response · Authors · 2024-11-26
>
> Weaknesses:
>
> W1: We emphasize that the LePP is only a variation of our proposed framework SKT on prompt-based CL algorithm. In addition, the application of transferability estimation in NLP is underexplored, and how to effectively apply it in continual learning with NLP tasks is overlooked, we paved the way for further investigation. We also demonstrate that it can help to understand the task relation during training, which is valuable for practitioners to gain their understanding. In contrast, other methods require significant computational overhead and the performance is sensitive to hyperparameters. Therefore, despite the simplicity of our proposed framework, we made a major contribution to the CL literature as highlighted in our contributions.
>
> W2: Please see the Q1 for our rebuttal.
>
> W3: We thank the reviewer for helping us improve the manuscript. We will carefully revise our manuscript and make changes for better understanding.
>
> Questions:
>
> Q1: We thank the reviewer for your feedback and updated the manuscript accordingly. However, we disagree with the reviewer that T5 is not always considered a LLM since T5 is the earlier work on LLMs. Although T5 is smaller than current LLMs such as LLaMa, GPT, and so on, the encoder-decoder architecture is still relevant today as it is used in SOTA models such as the REKA series (check reka.ai).
>
> Q2: Although decoder-only LLMs are capable of adapting to environments with ICL, they only can effectively work with environments that are close to the knowledge they trained on. However, LLMs still find it hard to adapt even with the support of ICL and still require continual training to acquire new knowledge and update facts  [1]. Therefore, efficient adaptation is still required due to the model complexity of current LLMs.
>
> Q3: We added more details to clarify the context of this research question. In particular, “forward transfer” is changed to “encouraging forward transfer” as how to effectively and efficiently use past knowledge to leverage new task learning is still a question in the literature. Other methods could be effective but not efficient. Our work aims to attain both criteria.
>
> Q4. We add more information to the tables to make it standalone. Please see the newest version for our changes.
>
> Q5. We conducted an additional experiment with the ViT experiment to empirically verify our claim in Section 4.4 in the main paper. We report our results in Appendix A.5 in the updated version. Using LePP only takes 52s in the total running time, equivalent to 0.6% additional computational overhead. Noting that there are 500 samples per class in this setting, therefore in a fewer-shot setting, the computational overhead is minimal.

---

> > ### Comment · Reviewer_FhGR · 2024-12-02
> >
> > Good to see the improvements. I increase my score to 5.

---

### Official Review · Reviewer_EiHR · 2024-11-06

**Soundness:** 3
**Presentation:** 2
**Contribution:** 2
**Rating:** 5
**Confidence:** 3

**Summary:**

This paper focuses on few-shot continual learning for large language models and introduces the Selective Knowledge Transfer (SKT) framework. The framework leverages transferability measures to autonomously identify relevant past memories for current tasks, enhancing knowledge transfer and improving system performance. The paper integrates SKT into existing prompt-based continual learning algorithm, Progressive Prompts, which learns a prompt for each new task. The paper empirically shows the effectiveness of SKT for BERT and T5 models on various NLP benchmarks. Additionally, SKT is shown to work with various data modalities, including images.

**Strengths:**

1. The proposed approach is straightforward and computationally efficient.
2. The Selective Knowledge Transfer framework effectively facilitates knowledge transfer between tasks.
3. The performance of LePP surpasses baseline methods on several continual learning benchmarks.

**Weaknesses:**

1. The writing of this paper could be improved. There are a few writing errors, e.g. line 65 and 168-169, and some areas that are confusing, e.g. the baseline and corresponding reference in line 262 and 294.
2. The paper assumes that task-ids are available during inference, allowing the framework to use a specific prompt for each task. However, obtaining these identifiers in real-world scenarios may be challenging.
3. It would be beneficial to include comparisons with more recent approaches, such as SAPT: A Shared Attention Framework for Parameter-Efficient Continual Learning of Large Language Models (ACL 2024) and Mitigate Negative Transfer with Similarity Heuristic Lifelong Prompt Tuning (ACL 2024).
4. The experimental results do not clearly demonstrate the effectiveness of positive knowledge transfer. The results only provide average performance of the models. It is important to compare continual learning methods from different perspectives to demonstrate positive knowledge transfer in a more direct way.
5. While the authors claim that the proposed framework can be applied to any Parameter-Efficient Tuning method, only results for Prompt Tuning are provided.
6. The proposed method is evaluated only on classification task, while its effectiveness on other generation tasks remains unclear.

**Questions:**

1. The largest language model used in the experiments is T5-large(770M), which may not be sufficient to demonstrate scalability with the size of language models. Have the authors experimented with larger language models, e.g. Llama3-8B?
2. Could the authors explain how the proposed framework can be applied to other PEFT methods?
3. How does the method work if task-ids are not available during inference?

---

> ### Author Response · Authors · 2024-11-26
>
> W1: We updated our manuscript accordingly. We thank the reviewer for pointing out our mistakes.
>
> W2: Please see the Q3 for our response.
>
> W3: We highlight that those two methods are less computationally efficient as they use training-based procedures for similarity calculation. In addition, they employ the linear maps [1,2] to project the soft prompt to the text embedding space, incurring additional parameters to the models. As a result, these methods are sensitive to hyper-parameters and require significant tuning effort for better outcomes. In contrast, our approach is training-free and principled by Bayesian model selection (LogME). Therefore, our proposed method is more computationally efficient by easing additional tuning efforts and is robust to sample size. We will provide additional experiments with those two methods in the camera-ready version.
>
> W4: We further provide the per-task performance for reference. By optimizing per-task learning, we can obtain the overall system performance. We further highlight that we use the same metrics as previous studies [1,2].
>
> W5: Although such extensions are beyond the scope of this work, we demonstrate how to apply our framework with LoRA in the answer Q2.
>
> W6: We only evaluated classification tasks based on the benchmark from Progressive Prompt. However, our proposed framework can be applied to various tasks. For instance, given a specific length of generated text for generation tasks, we can consider those tasks as high-dimensional regression tasks and LogME is still applied. We also provide additional experiments in the final version of this paper.
>
> Questions:
>
> Q1: Due to our tight computational budget, we only conducted experiments with T5-based models. However, using T5-based models is equivalent to prior works including Progressive Prompt.
>
> Q2: Our proposed framework could be applied to other weight incremental methods such as LoRA and Adapters. For instance, we assume that each task has different residual weights (e.g. LoRA module). First, we merge the task weight incremental with the base model to extract the task’s feature, which is then employed to calculate the transferability score a.k.a the importance weight of each task.
>
> Q3: When task IDs are unavailable, we can still apply the key-query mechanism introduced in Dual-Prompt to find the task ID corresponding to an input. However, in this case, our method suffers from the task-id-prediction error as the key-based CL algorithm.
>
> References:
>
> 1. Progressive prompts: Continual learning for language models, ICLR 2023.
> 2. Orthogonal Subspace Learning for Language Model Continual Learning, EMNLP 2023 Findings.

---

> ### Author Response · Authors · 2024-12-02
>
> Dear reviewer EiHR,
>
> This is a reminder that the rebuttal round will end today. Please review our response and increase your rating if it addresses your concerns. If you have any further questions, please let me know.
>
> Best regards.

---

### Meta-Review · Area_Chair_5jch · 2024-12-16

**Metareview:**

(a) Summary

This paper investigates how to aggregate knowledge from previous tasks in continual few-shot learning settings. It proposes a selective knowledge transfer(SKT) framework for enhancing positive knowledge transfer while reducing the interference from irrelevant tasks. Furthermore, it introduces Log-evidence Progressive Prompts (LePP) for integrating the proposed framework to continual LLM learning setting. Experiments validate the effectiveness of the proposed method.

(b) Strengths
+ The proposed method is both simple and effective for combining knowledge transfer and CL.
+ Several experiments have been conducted to demonstrate the effectiveness of the proposed method.
+ LePP performs better than baselines on CL benchmarks.

(c) Weaknesses
- The assumption is not practical: it assumes task ids are available during inference time.
- It has limited novelty: The proposed method LePP heavily relies on previous work LogME and Progressive Promt.
- It has limited application: LePP cannot be directly applied to decoder-only models which are the majority of "LLMs".
- The experiments do not show positive knowledge transfer but average transfer.
- The experiments conver limited task type and model type: small dataset for classification tasks.  Its effectiveness on other generation tasks remains unclear. It would be also beneficial to include comparisons with more recent approaches, such as SAPT.

(d) Decision

This paper proposed a simple and effective method to perform knowledge aggregation in continual language learning. However, as pointed out by the reviewers, there are limited novelty and limited real-world applications due to the impractical assumption.  For these reasons, I recommend reject.

**Additional Comments On Reviewer Discussion:**

This is a borderline paper. The reviewers shared the concerns on the limited novelty of the proposed LePP method, its assumption on available task ids during inference time, missing experimental settings, and its limited practical applications. The rebuttal and updated manuscript have addressed part of the reviewers' concerns, but these were not enough to sway reviewers for positive scores.

---

### Decision · Program_Chairs · 2025-01-22

Reject